# Development of a Broad-Spectrum Pan-Mpox Vaccine via Immunoinformatic Approaches

**DOI:** 10.3390/ijms26157210

**Published:** 2025-07-25

**Authors:** Japigorn Puagsopa, Panuwid Jumpalee, Sittichoke Dechanun, Sukanya Choengchalad, Pana Lohasupthawee, Thanawat Sutjaritvorakul, Bunyarit Meksiriporn

**Affiliations:** 1Department of Physiology and Aging, College of Medicine, University of Florida, Gainesville, FL 32610, USA; jpuagsopa@ufl.edu; 2Department of Biology, School of Science, King Mongkut’s Institute of Technology Ladkrabang, Bangkok 10520, Thailand; panuwid.work@gmail.com (P.J.); sukanya.choeng@gmail.com (S.C.); panaloh@gmail.com (P.L.); 3Department of Bioengineering BioScience Research Collaborative, Rice University, Houston, TX 77005, USA; sd123@rice.edu; 4Faculty of Science and Technology, Pathumwan Institute of Technology, Bangkok 10330, Thailand; thana5306@hotmail.com

**Keywords:** Mpox, immunoinformatics approach, broad-spectrum vaccine, molecular docking, molecular dynamics (MD) simulations

## Abstract

Monkeypox virus (MPXV) has caused 148,892 confirmed cases and 341 deaths from 137 countries worldwide, as reported by the World Health Organization (WHO), highlighting the urgent need for effective vaccines to prevent the spread of MPXV. Traditional vaccine development is low-throughput, expensive, time consuming, and susceptible to reversion to virulence. Alternatively, a reverse vaccinology approach offers a rapid, efficient, and safer alternative for MPXV vaccine design. Here, MPXV proteins associated with viral infection were analyzed for immunogenic epitopes to design multi-epitope vaccines based on B-cell, CD4+, and CD8+ epitopes. Epitopes were selected based on allergenicity, antigenicity, and toxicity parameters. The prioritized epitopes were then combined via peptide linkers and N-terminally fused to various protein adjuvants, including PADRE, beta-defensin 3, 50S ribosomal protein L7/12, RS-09, and the cholera toxin B subunit (CTB). All vaccine constructs were computationally validated for physicochemical properties, antigenicity, allergenicity, safety, solubility, and structural stability. The three-dimensional structure of the selected construct was also predicted. Moreover, molecular docking and molecular dynamics (MD) simulations between the vaccine and the TLR-4 immune receptor demonstrated a strong and stable interaction. The vaccine construct was codon-optimized for high expression in the *E. coli* and was finally cloned in silico into the pET21a (+) vector. Collectively, these results could represent innovative tools for vaccine formulation against MPXV and be transformative for other infectious diseases.

## 1. Introduction

Monkeypox is a zoonotic disease caused by monkeypox virus (MPXV), which is an enveloped double-stranded DNA (dsDNA) virus that is categorized in the Orthopoxvirus genus of the Poxviridae family. The symptoms of monkeypox include fever, rash, lymphadenopathy, and intense asthenia; however, certain groups (children, pregnant women, and immunocompromised people) might suffer from serious complications, including secondary bacterial infections, pneumonitis, respiratory distress, sepsis, encephalitis, and loss of vision due to corneal infection. The fatality rate of MPX ranges between 0 and 11%, with a self-limiting disease lasting 2–4 weeks [1]. MPXV was first discovered in monkeys in 1970; however, it was first confirmed in humans in 1970 in a 9-month-old child with suspected smallpox in the Democratic Republic of the Congo [2]. Thereafter, infection cases have been occasionally reported, but most MPX cases were restricted to West African and Central African nations, with a mortality rate between 1 and 10% [3,4]. Monkeypox was documented to be transmitted by travelers from human-to-human transmission or by exposure to imported animals [5]. Since 2022, the global incidence of monkeypox (MPX) disease caused by the virus has dramatically increased worldwide, with over 148,892 laboratory-confirmed infection cases and more than 341 deaths, reported by the World Health Organization (WHO), from 137 reporting countries (as updated in WHO website on 31 May 2025) [6]. Given the widespread prevalence of MPX disease, the WHO declared this disease a public health emergency of international concern in July 2022. Due to its genetic similarity to smallpox infection, patients with severe MPXV infection are treated with antiviral medicines for smallpox virus infection, including tecovirimat, cidofovir, and brincidofovir, which are currently used to treat MPX under certain conditions; however, high-dose antiviral drug medication tends to cause hepatic malfunctions, while low-dose treatment may lead to the risk of relapse [7]. Thus, there is an urgent demand for effective vaccines against MPXV infection to curb the number of cases.

However, generating such vaccines remains a significant challenge, and to date, no specialized vaccines have been developed specifically to protect against MPX infection. Several smallpox vaccines, including ACM2000, MVA-BN, and LC16, are available for MPX vaccination in different jurisdictions. The US FDA-approved ACAM2000 is available for smallpox and MPX vaccination. However, ACAM2000 is a live, replication-competent vaccinia virus, and there is a risk of side effects such as injection site pain, lymph node pain, pruritus, and other flu-like symptoms. Additionally, serious and long-term side effects, including myopericarditis or pericarditis, have been reported in every 20 cases out of 100,000 ACAM2000 vaccine recipients [8]. MVA-BN, also known as JYNNEOS, a nonreplicating modified Vaccinia Ankara virus vaccine, was approved by the US FDA in 2019 for the prevention of both smallpox and monkeypox infections in the U.S. Since Jynneos is a replication-deficient live virus vaccine, it is safer than ACAM2000, particularly for vaccination in immunocompromised individuals, such as transplant recipients, patients with HIV, and atopic dermatitis [9,10]. However, limited data on the efficacy of Jynneos against MPXV infection in humans have been documented [11]. LC16, a third-generation smallpox vaccine, is a live, replicating attenuated vaccine derived from the Lister (Elstree, UK) strain of vaccinia that was approved for the prevention of MPX in August 2000 in Japan [12,13]. Although preclinical evidence has demonstrated a safe profile in animal models, the LC16 vaccine is not recommended for administration to immunocompromised patients or atopic dermatitis patients [8,14,15]. Nevertheless, the efficacy of these vaccines against monkeypox is contingent on data derived from animal studies demonstrating immune protection against the monkeypox virus in nonhuman primates rather than human subjects [16].

The mutation rate of MPXV was 6–12-fold greater than previously expected, and some of these mutations led to increased transmissibility and immune evasion. Thus, there is an urgent need for novel vaccine design methods [12,17]. The drawbacks of conventional vaccine development include low throughput, high cost, and high time consumption, as well as the need for pathogen culture and the identification of immunogenic components [18,19]. To address these shortcomings in conventional vaccine development, reverse vaccination approaches have revolutionized vaccine design, thus enabling scientists to identify immunogenic components and highly conserved epitopes from genome sequence and proteome information without the need to isolate and culture the targeted pathogen and, thus, offering time and resource savings [20].

Multi-epitope vaccines developed by immunoinformatic approaches have demonstrated robust immune protection against microbial infection because of fewer side effects, greater specificity, greater safety, lower-cost production than conventional vaccines, and the simultaneous induction of diverse immune responses, including innate, cellular, and humoral immune responses [21,22].

The critical step in developing an effective multi-epitope vaccine is target selection. Since there are no reports on specific MPXV proteins that are predominantly involved in MPXV infection, several proteins, including structural and nonstructural proteins, could play critical roles in MPXV infection and could be promising targets for vaccine development. Several MPXV proteins, including M1R (myristylprotein), B6R (EEV type-I membrane glycoprotein), and H3L (IMV heparin binding surface protein), are associated with immune responses within poxviruses and are considered attractive targets for vaccine development [23]. Additionally, B9R has been implicated in the pathogenesis of MPXV [24], whereas F8L (DNA polymerase) has been reported to stimulate the CD8+ T-cell population and the release of interferon-g in rhesus macaques [25]. Notably, a previous study demonstrated the use of M1R and B6R as components of a recombinant protein vaccine, and the results showed significant inhibition of MPXV in mice [26]. Accordingly, both structural proteins and nonstructural proteins could be promising targets for vaccine development.

Here, a panel of multi-epitope vaccines was designed against MPXV using a reverse vaccination approach. Figure 1 illustrates the overall schematic of this study. The MPXV proteins were retrieved from the GenBank database and filtered through several criteria to select final candidates for vaccine construction. Both structural proteins and nonstructural proteins were selected as protein targets for multi-epitope vaccine development. According to previous studies, B6R, H3L, and F8L can stimulate interferon-γ production and activate cytotoxic T lymphocytes (CTLs) and helper T lymphocytes (HTLs) [26,27,28,29,30]. E4R has been reported to induce antibodies in vivo in prairie dogs [31]. A40L, B9R, and C2L are homologous to the A38L, B8R, and K2L vaccinia viral proteins, which are reportedly crucial ligands for MHC class II and HTL [32,33]. J3R/J1l plays a pivotal role in immunomodulating and preventing apoptosis in host cells, thus facilitating viral entry and infection [34]. Moreover, to provide broad-spectrum protection, six different MPXV strains were used as our research objects. All candidate epitopes were then further selected based on their antigenicity, low allergenicity, and low toxicity. To enhance immunostimulation, an adjuvant protein sequence was fused to the N-terminus of each vaccine construct. Overall, our molecular docking and molecular dynamics simulation data suggested that our vaccine candidate could bind to and trigger an immune response via TLR-4. Hence, our MPXV multi-epitope vaccine construct could be a viable alternative for preventing the global outbreak of the MPX pandemic.

## 2. Results

### 2.1. Monkeypox Viral Protein Retrieval and Screening

Monkeypox protein sequences, including structural and nonstructural proteins retrieved from the MPXV strain Congo_2003_358 in the NCBI database (DQ011154.1), were predicted for antigenicity and allergenicity via VaxiJen 2.0 and AllergenFP1.0. From the 200 proteins retrieved from the completed genome, only 24 proteins were selected based on non-allergenicity and antigenicity, with a score higher than 0.4, as demonstrated in Table 1. These 24 selected proteins were further utilized for epitope prediction.

### 2.2. Prediction of Epitope Candidates and Assessment of Allergenicity, Antigenicity, and Toxicity Parameters

The NetMHCpan 4.1 BA and NetMHCpan 4.1 EL servers predicted 35 CTL epitopes that strongly bind to MHC class I for HLA class I alleles (HLA-A0101, HLA-A0201, HLA-A0301, HLA-A2402, HLA-A2601, HLA-B0702, HLA-B0801, HLA-B2705, HLA-B3901, HLA-B4001, and HLA-B5801). All 35 selected CTL epitopes demonstrated a positive immunogenicity score via the IEDB Class I immunogenicity tool, antigenicity with a score higher than 0.4, non-allergenicity, and non-toxicity.

HTL epitopes were predicted through NetMHCpanII 4.1 BA and NetMHCpanII 4.1 EL for MHC class II. These servers predicted 24 potential HTL epitopes with strong binding affinities against MHC class II based on the HLA alleles DRB1_0101, DRB1_0301, DRB1_0401, DRB1_0701, DRB1_0801, DRB1_0901, DRB1_1001, DRB1_1101, DRB1_1201, DRB1_1301, DRB1_1501, and DRB1_1602. Additionally, these HTL epitope candidates were tested with the IFNepitope server to selectively filter those sequences that can stimulate IFN-γ. All 24 HTL epitopes could trigger IFN-γ secretion. Out of the 24 epitopes, only one epitope had a low antigenicity score (less than 0.4); thus, 23 of the remaining HTL epitopes had high antigenicity, no allergenicity, and no toxicity.

In the case of LBL, the BepiPred linear epitope prediction 2.0 server was used to predict the LBL epitopes with the default threshold of 0.5. The server yielded a total of 30 epitopes predicted to have high antigenic scores, no allergenicity, and no toxicity. The CTL, HTL, and LBL epitope repertoires were utilized for further analysis to select final candidates for vaccine construction.

### 2.3. Transmembrane and Signal Peptide Screening

The CTL, HTL, and LBL predicted epitopes obtained from previous experiments were further screened for transmembrane regions and signal peptides by utilizing DeepTMHMM 2.0 and SignalP 6.0. DeepTMHMM 2.0 was used to predict the transmembrane regions of epitopes that should be removed to avoid hydrophobic effects that are prone to aggregation, thus leading to low yields and unstable protein synthesis [8,35,36] (Appendix A Figure A1, Figure A2, Figure A3, Figure A4, Figure A5, Figure A6, Figure A7, Figure A8, Figure A9 and Figure A10). The DeepTMHMM 2.0 platform has demonstrated superior performance compared to other methods across most protein types [37]. Furthermore, SignalP 6.0 was evaluated in comparison with existing prediction tools, which, in contrast to SignalP 6.0, lack the capability to identify all classes of signal proteins. Additionally, the outcomes from these tools exhibited inherent biases, as they could not be assessed within a cross-validation framework [38]. The CTL, HTL, and LBL epitopes demonstrating potential transmembrane regions were retracted from the constructed vaccines. Our CTL, HTL, and LBL epitopes contained no transmembrane region. Finally, SignalP 6.0 was used to predict the signal peptide regions of the CTL, HTL, and LBL epitopes, and no signal peptides were found in our CTL, HTL, or LBL epitopes.

### 2.4. Epitope Conservancy for the Broad-Spectrum Coverage of Vaccines Across Different Clades

To generate broad-spectrum vaccines against various clades of Monkeypox virus, all the predicted CTL, HTL, and LBL epitopes were analyzed for epitope conservation across seven clades of Monkeypox virus. Nearly all the epitopes covered Clade I (Zaire-96-l-16), Clade IIa (USA 2003), Clade llb-A (Nigeria 2018), Clade IIb-A.2 (USA 2020), Clade IIb-A.1 (UK P2), Clade IIb-A.1.1 (USA2021), and Clade IIb-B.1 (France 2022), with 100% conservancy. However, an HTL epitope derived from the F8L protein (SMVFEYRASTVIKGP) showed only a single amino acid variation from valine to isoleucine. One valine in Clade I, Congo_2003_358 and Zaire-96-l-16 was replaced by an isoleucine in Clade IIa (USA 2003), Clade llb-A (Nigeria 2018), Clade IIb-A.2 (USA 2020), Clade IIb-A.1 (UK P2), Clade IIb-A.1.1 (USA2021), and Clade IIb-B.1 (France 2022). In other words, the SMVFEYRASTVIKGP in Clade I became the SMVFEYRASTIIKGP in Clade II (Appendix A Table A1). Considering the amino acid properties, both valine and isoleucine are categorized as hydrophobic amino acid groups; thus, the valine to isoleucine substitution should have a negligible effect on the epitope.

### 2.5. Epitope Homology Analysis with the Human Proteome

To prevent autoimmune induction by epitopes, the Peptide Search server from the UniProt database was used to investigate epitopes that showed homologous sequences to the human proteome. According to the peptide match tool, the predicted epitopes were checked against the Homo sapiens proteome (Taxonomy ID: 9606). The results showed that all the epitopes were non-homologous to the human proteome.

### 2.6. Final Epitope Selection and Worldwide Population Coverage

Epitopes showing nonhomologous sequences to the human proteome were further selected for the vaccine construct. To ensure broad HLA coverage, epitopes were chosen to target different alleles for both CTL and HTL. These candidates were selectively recruited based on HLA allele coverage for CTL and HTL epitopes and high antigenicity for all three types (CTL, HTL, and LBL). Following this method, 9 CTL, 4 HTL, and 4 LBL epitopes were selected, as illustrated in Table 2a–c.

Afterwards, the selected CTL and HTL epitopes were analyzed via the IEDB population coverage tool. The distribution of HLA alleles was targeted by the combination of the predicted MHC class I and II epitopes. The results showed that 94.87% of the world population across 115 nations and 21 ethnicities was covered within 16 geographic regions, as shown in Figure 2.

### 2.7. Multi-Epitope Subunit Vaccine Construct

The design of the vaccine construct was formulated by combining five elements, including an adjuvant, nine CTL epitopes, four HTL epitopes, four LBL epitopes, and a 6xHis tag, with the help of linkers, as illustrated in Figure 3. Linkers have been reported to provide stability to vaccine constructs and prevent cross-interruption of each epitope, thus allowing them to independently function upon vaccination [39,40].

Incorporating the adjuvants into the vaccine construct could improve the immune response. Therefore, in this study we selected five different adjuvants, namely, cholera toxin B subunit (CTB), PADRE, beta-defensin 3, 50S ribosomal protein L7/12, and RS-09. The study showed that CTB can enhance immune responses by binding to GM1 ganglioside receptors on various immune cells [41]. PADRE elicits stronger antibody responses compared to traditional T-cell epitopes [42,43]. Adjuvants like RS09 can improve vaccine efficacy by enhancing, modulating, and prolonging immune responses, while reducing antigen concentration and required immunizations [44]. The 50S Ribosomal L7/L12 can enhance dendritic cell maturation, promote Th1 polarization, and improve T-cell-mediated cytotoxicity, leading to better tumor immunotherapy outcomes [45]. Beta-defensins offers several advantages. They enhance both humoral and cell-mediated immune responses, leading to improved vaccine efficacy against various pathogens, including influenza, MERS-CoV, and IBDV [46,47,48].

These adjuvants were N-terminally fused to the vaccine by using the EAAAK linker. The CTL epitopes were linked with AAY linkers, and GPGPG linkers were used to connect the HTL epitopes. The B-cell epitopes were linked with the help of KK linkers. HEYGAEALERAG linkers were used to connect the first HTL epitope and the last CTL epitope as well as to link the first B-cell epitope to the last HTL epitope. Additionally, the 6xHis tag was fused to the C-terminus of the vaccine construct by the RVRR linker for vaccine purification by affinity chromatography and immunodetection of vaccine expression. A schematic of each vaccine construct is depicted in Figure 3. The antigenicity, allergenicity, toxicity, solubility, presence of signal peptide, and physicochemical properties of all five vaccine constructs were predicted and compared to those of a control vaccine without an adjuvant.

### 2.8. Antigenicity, Allergenicity, Toxicity, and Physicochemical Properties

The five vaccine constructs were evaluated and compared for antigenicity, allergenicity, toxicity, and physicochemical properties, as shown in Table 3. All five vaccine constructs demonstrated antigenicity scores higher than 0.50 according to the VaxiJen 2.0 server, which indicated the robust antigenic properties of all the multi-epitope vaccine constructs. Additionally, four vaccines were immunogenic and nontoxic; however, the beta-defensin adjuvant was toxic. The physicochemical properties predicted from the ProtParam server showed that the molecular weights of all the multi-epitope vaccine constructs ranged from ~29 to ~40 kDa. The theoretical isoelectric points (PIs) of all the constructs ranged from 5.46 to 8.71. The estimated half-lives of in vitro mammalian reticulocytes were 30 h, >20 h in in vivo yeasts, and >10 h in in vivo *E. coli*. The instability indices of the cholera toxin B-vaccine, PADRE vaccine, RS09-vaccine, beta-defensin-vaccine, and 50S ribosomal protein L7/12 ranged from ~29 to ~37, which indicated that these constructs were stable at various temperatures [49]. The gravy scores of all the vaccines ranged from −0.427 to −0.699, indicating a hydrophilic nature [49,50]. In line with the Gravy scores, the solubility scores calculated by the SOLpro server ranged from ~0.72 to ~0.98, indicating high solubility [35]. The aliphatic indices of cholera-vaccine, PADRE vaccine, the RS09-vaccine, and the 50S ribosomal protein L7/12 ranged from 65.88 to 77.81, indicating high thermostability, whereas the aliphatic index of the beta-defensin-vaccine was 35.21, indicating low thermostability [36]. Finally, the signal peptide sequence of the vaccine was also tested; however, no signal peptide sequence was found within any of the tested vaccines. Finally, the signal peptide sequence of the vaccine was also tested; however, no signal peptide sequence was found within any of the tested vaccines. Collectively, based on the above predictions, the PADRE vaccine demonstrated overall improvement in terms of antigenicity, solubility, and stability without compromising allergenicity or toxicity. Thus, the PADRE vaccine was recognized as the lead vaccine construct for immunological translation and was chosen for further analysis.

### 2.9. Secondary Structure Prediction, Three-Dimensional Structure Predictions, Refinement, and Validation

The PADRE vaccine showed overall improvements in both immunological and physicochemical properties; therefore, it was used to predict secondary and three-dimensional structures. The PSIPRED 4.0 server predicted our vaccine construct to include 44.6% coils, 33.7% α-helices, and 21.8% β-strands, as shown in Figure 4. In addition, we used Alphafold 3 to generate the three-dimensional structure models, in which the top five with the highest pLDDT was selected for further analysis. To improve the accuracy of the selected structure model, we further refined by using the GalaxyWEB server, resulting in models with improved quality, as demonstrated by quality assessment parameters, as shown in Table 4. Model 1 was selected based on improved Molprobity and Ramachandran plot favored scores of 1.302 and 98.2, respectively, which indicated the high quality of the model. Pymol was used to visualize the 3D model of the vaccine construct (Figure 5).

Model 1 was further validated with the SAVES 6.0 server by ERRAT and PROCHECK. The selected model exhibited a quality score of 78.7149% in accordance with ERRAT. According to the plot of PROCHECK for evaluation via Ramachandran, 94.7% of the residues were found in the preferred region, and 5.3% were found in the allowed region, as shown in Figure 6a. Additionally, the ProSa-Web server was used to compute the z-score of the model. The z-score calculated by the ProSa-web server was −6.42, as shown in Figure 6b. Collectively, the results from ERRAT, PROCHECK, and ProSa-Web indicated the excellent quality of the predicted three-dimensional model.

### 2.10. Molecular Docking of TLRs and Selected Vaccine Constructs

The ClusPro 2.0 server was used to provide the protein–protein docking. The results included cluster size, center energy score, and lowest energy score. Among the 30 best-fit docking models, the model with the largest cluster size was prioritized for further analysis. This model yielded a score of −1242.8 for the CES and −1438.3 for the lowest energy consumption, as shown in Table 4. The lead model was refined via the REFINEMENT tool on the HADDOCK 2.4 server. Subsequently, the model was analyzed for Gibb’s free energy (ΔG) and dissociation constant (K_D_), as well as other aspects, including the HADDOCK score, cluster size, RMSD from the overall lowest-energy structure, van der Waals energy, electrostatic energy, desolvation energy, restraint violation energy, buried surface area, and z-score, as shown in Table 4. The predicted model of the vaccine-TLR4 complex was selected. The binding energy (ΔG) of this model was −19.3 kcal/mol, and the dissociation constant (K_D_) was 2.3 × 10^−14^ M. With respect to the van der Waals energy, electrostatic energy, desolvation energy, restraint violation energy, and buried surface area, these parameters contributed to a HADDOCK score of −568.2 ± 9.5 for the vaccine-TLR4 complex. Additionally, the cluster size of the complex was 20. The root mean square deviation (RMSD) from the overall lowest-energy structure was 0.6 ± 0.3. Moreover, a z-score of 0 showed that this model was exactly the average score within the cluster. Overall, these parameters indicated that the selected model was stable and energetically favorable.

### 2.11. In Silico Immune Stimulation

The C-IMMSIM v10.1 server was used to predict the immunization efficacy of the candidate vaccine. The results from stimulation using the candidate vaccine included B cells, helper T cells, cytotoxic T cells, natural killer (NK) cells, macrophages (MAs), dendritic cells (DCs), and certain cytokines. The PADRE vaccine triggered IgM and IgG production after primary immunization and elevated the production of IgM, IgG1, and IgG2 after the secondary and tertiary responses, as shown in Figure 7a,c. In line with the elevated antibody titer, total B-cell populations, particularly B isotype IgM and memory B-cell populations, were induced after the first injection (Figure 7b). The total B cell, IgM, memory B cell, and IgG1 populations were dramatically elevated, while the B isotype IgG2 was also induced after the second and third injections, as shown in Figure 7b. Noticeably, a decrease in the antigen concentration was detected, as shown in Figure 7a, and these phenomena resulted from the generation of a memory B-cell population (Figure 7b). Additionally, Figure 7c indicates that the plasma B lymphocyte count (isotype: IgM + IgG) was noticeably elevated. The active B-cell population per state increased significantly and became consistently high after immunization, as demonstrated in Figure 7d. According to Figure 7e, the total helper T-cell population increased significantly after the first, second, and third injections. In particular, both memory and nonmemory helper T cells were elevated. Likewise, the active and resting helper T-cell populations per state could be elevated significantly, as indicated by Figure 7f. Although the resting cytotoxic T-cell population per state decreased after the first immune response, it dramatically increased over time after the tertiary immune response, as shown in Figure 7g. Total NK cells, the total MA population per state, especially the resting MA population, and the DC population per state (for both active and resting states) were also generated by immunization, as demonstrated in Figure 7h–j. As shown in Figure 7k, the titer of interferon-γ (IFN-γ) was significantly increased by all three injections. Increased transforming growth factor-β (TGF-β), interleukin 10 (IL-10), and interleukin 12 (IL-12) levels were observed. An elevated danger level D in the same figure indicates that the risk is extremely low. This represented a favorable reaction to immunization.

### 2.12. Codon Optimization and In Silico Cloning

Codon optimization was performed using the Java Codon Adaptation Tool (JCAT) for protein expression in *E. coli*. The protein sequence of the PADRE vaccine was reverse-translated to the DNA sequence. The DNA sequence was submitted to the JCAT server for codon optimization. The codon adaptation index (CAI) and GC content of the codon-optimized sequence were 1.0 and 51.11%, respectively (Appendix A Figure A11). These scores indicated that the optimized nucleotide sequence was within the range of the optimum CAI (0.8–1.0) and GC content (30–70). Finally, the optimized nucleotide sequence above was in silico cloned by insertion into the pET21a vector digested with the NdeI and XhoI restriction enzymes using SnapGene 7.2.1, resulting in the recombinant plasmid pET-21a(+)-PADRE Mpox vaccine-6xHis tag, as shown in Figure 8.

### 2.13. NMA via the iMODS Server

The iMODS server and GROMACS software were utilized to perform molecular dynamics (MD) simulations. iMODS was used to analyze the vaccine-TLR complexes via normal mode analysis (NMA) to investigate the binding affinity of the vaccine-TLR4 complex and to investigate its stability and dynamic movements over time. The main-chain deformity was determined by measuring the extent of molecular deformation at each residue. High peaks represented the deformable regions of the vaccine-TLR complex, indicating flexible locations, while rigid regions were indicated by low values, as shown in Figure 9. According to Figure 9a, the vaccine-TLR4 complex demonstrated low distortion.

B-factor charts, shown in Figure 9b, indicate the average RMSD values of the vaccine-TLR complex obtained by relating the mobility of the computed complex in NMA to the corresponding PDB field. The eigenvalues of the docked materials, illustrated in Figure 9c, explained the motion stiffness. The required energy for structural deformation was indicated by the eigenvalue, with a lower eigenvalue representing easier deformation. The eigenvalue of the interaction between the vaccine and TLR4 was 6.239649 × 10^−7^. According to Figure 9d, the graphs of variance, which showed an inverse association with the eigenvalue, are shown. The purple color represents individual variance, while the green color represents cumulative variance for each normal mode of the vaccine-TLR complex. The covariance map of the vaccine-TLR complex illustrated coupling between two molecules in the complex (Figure 9e). Red represents correlated atomic motions, white represents uncorrelated atomic motion, and blue represents anticorrelated atomic motion in the regions of MD depicted in Figure 9e. The elastic network map of the vaccine-TLR4 complex shown in Figure 9f depicts each gray dot as the formation of springs connecting the corresponding pair of atoms. Flexible regions are represented by lighter dots, whereas stiffer regions are indicated by darker dots. Collectively, these results highlighted the stable interaction between the vaccine and TLR4.

### 2.14. MD Analysis via GROMACS

To measure the stability of the vaccine-TLR4 complex, the backbone atom RMSD was computed from the initial point in time. According to Figure 10a, the RMSD of the complex rapidly increased from the beginning to approximately 5 ns. After that, the structure stabilized at approximately 20 ns, within the range of approximately 0.6 to 0.5 nm. However, there was an increase after 20 ns until it became substantially stable after approximately 24 ns, with a range of approximately 0.6–0.8 nm throughout the remaining simulation time. Additionally, the RMSF was analyzed to assess the flexibility of the vaccine-TLR complex. The high fluctuation in the RMSF indicated highly flexible regions of the complex. The peak corresponding to residue 636 was at 2.1589 nm, as shown in Figure 10b. Several highly fluctuating regions were identified, including the region between residues 409 and 519 (with a range of 0.8203 to 1.287 nm), the region between residues 996 and 1222 (with a range of 0.7031 to 1.2779 nm), the region between residues 628 and 653 (with a range of 1.0358 to 2.1589 nm), and the region between residues 1267 and 1558, with a very wide range of fluctuations between 0.2887 and 1.5996 nm. The last region contained several highly fluctuating residues, such as residue 1267 (with a length of 1.5996 nm), residue 1414 (with a length of 1.4612 nm), and residue 1558 (with a length of 1.5258 nm).

Radius of gyration (Rg) was analyzed to determine the structural compactness of the vaccine-TLR complex. As illustrated in Figure 10c, the overall Rg was substantially stable throughout the simulation, and did not change abruptly over time. Furthermore, the stability of the complex could be measured by the number of hydrogen bonds. According to Figure 10d, the results of the MD analysis illustrated the stability of the number of hydrogen bonds within a very narrow range between 10 and 11 throughout the simulation time.

Finally, the SASA could indicate the surface area of a vaccine-TLR complex exposed to solvent molecules. These results could imply the flexibility and amenability of protein–protein interactions. As illustrated in Figure 10e, there was a fluctuation from the beginning of the simulation to approximately 20 ns, with a range of ~760 to ~720 nm^2^. After 20 ns until 34 ns, the SASA became more stable. Then, there was a sudden drop at approximately 36 ns; however, the SASA maintained its stability from ~36 to 42 ns. Eventually, it increased again at a simulation time of ~43 ns, after which the SASA became stable until the end of the simulation time.

## 3. Discussion

Monkeypox is a zoonotic double-stranded DNA virus in the *Orthopoxviral* genus. Recently, there has been a global outbreak of monkeypox disease, with 148,892 confirmed cases and 341 deaths worldwide according to WHO, as updated in WHO website on 31 May 2025 [6]. To address the global outbreak, several smallpox vaccines, including ACAM2000, JYNNEOS (MVA-BN), and LC16, have been introduced to offer cross-protection against monkeypox [8]. This also means that nonspecific vaccines are being used to provide protection against monkeypox infection. A small study using previous data from Africa showed that the smallpox vaccine might offer approximately 85% protection against monkeypox infection [51].

The Advisory Committee on Immunization Practices (ACIP) recommends the ACAM2000 live-attenuated vaccinia virus vaccine for specific personnel groups. These groups include laboratory personnel routinely handling human infectious *orthopox* viruses (monkeypox, cowpox, and variola) and medical personnel caring for vaccinia virus-infected patients [52]. However, ACAM2000 carries potential safety risks. Studies have reported occurrences of postvaccinal encephalitis, eczema vaccinatum, and progressive vaccinia following vaccination [52]. Additionally, the presence of live vaccinia virus raises concerns about secondary transmission to both vaccinated individuals and those in close contact and the potential for the development of myocarditis or pericarditis in some recipients [16]. Moreover, administration of the JYNNEOS vaccine has been associated with various side effects, including induration, chills, headache, sore throat, nausea, redness, myalgia, firmness/tightening, and pain [8]. A retrospective study conducted in a northwestern US population cohort (Oregon) identified 10 cases of cardiac events following JYNNEOS vaccination between July and October 2022 [53]. Similarly to JYNNEOS, LC16 is a live-attenuated, nonreplicating vaccine approved for monkeypox prevention in Japan in August 2022 [16]. Most adverse events were reported as mild to moderate localized or systemic in nature. Notably, clinical trials and cohort studies have not identified serious adverse events such as encephalitis or symptomatic myocarditis [8]. While preclinical data suggest potential safety for immunocompromised patients and atopic dermatitis patients, widespread vaccination in these populations is not currently recommended due to insufficient clinical data [8]. The observed limitations of live-attenuated vaccines suggest the need to explore alternative vaccination strategies that offer comparable efficacy with reduced adverse effects.

To address these shortcomings, a multi-epitope vaccine was proposed as a candidate for this study because, compared with traditional vaccines, this novel type of vaccine has a reduced capacity for eliciting pathological adverse effects due to the absence of unwanted components [54]. These unwanted components could be screened by using immunoinformatic tools. In tests with animal models and early clinical trials, multi-epitope vaccines induced cellular and humoral immune responses and elicited robust immune responses from individual epitopes against tumors and microbial and viral infections [54,55].

In this study, epitopes were recruited from monkeypox viral proteins, as sequences of both structural and nonstructural viral proteins were screened for only strong binding to the CTL, HTL, and LBL epitopes. Only strongly bound CTL epitopes that could induce class I immunogenicity and strongly bound HTLs that could induce interferon-γ were recruited for the vaccine construct. Moreover, the highest antigenic LBL epitopes were identified. These epitopes were also screened for allergenicity, toxicity, transmembrane regions, and human analogous proteomes. Among the 200 monkeypox viral proteins, epitopes were derived from eight proteins, namely, A40L, B6R, B9R, C2L, E4R, F8L, H3L, and J3R/J1L. These proteins were found to potentially induce the adaptive immune system and cytokines. In particular, previous studies have shown B6R and H3L to be immunodominant, as they stimulate CTLs, HTL, LBL, and interferon-γ [26,27,28,29,30]. The vaccinia viral proteins B5R and H3L, homologous proteins of the monkeypox viral protein B6R and H3L, respectively [56], were found to be immunodominant [32,33,57,58]. F8L is another highly dominant immunogenic protein that was found to trigger both HTL and CTL [25,59]. This protein was also tested in vivo and found to induce CTL in nonhuman primates [25]. E4R is another monkeypox viral protein that was reported to induce antibodies in vivo in prairie dogs [31]. Moreover, the monkeypox viral proteins A40L and C2L were not directly mentioned in previous studies. However, the vaccinia viral proteins A38L and K2L are homologous to A40L and C2L, respectively [56]. They were found to be crucial ligands for MHC class II and HTL [32,33]. B9R is homologous to the vaccinia viral protein B8R [56]. This protein was found to be immunodominant in CTL stimulation [60,61]. Finally, J3R/J1L, which are chemokine binding proteins, play critical roles in modulating and preventing apoptosis in host cells to facilitate viral infection [34]. The homologous vaccinia viral protein B29R of J3R/J1L was reported to be involved in acute and memory CTL stimulation with recombinant antigen [62].

The epitopes selected from the proteins above were investigated for epitope conservation. According to previous studies on vaccine design, there are still no vaccines that can provide broad-spectrum protection against all subclades. Even though the most virulent clade II, with fewer deadly effects, caused a global outbreak in 2022 with subclade IIb [63,64,65], highly fatal clade I was observed during the outbreak again from 2023–2024 [63,65]. Therefore, this study aimed to provide broad-spectrum protection from both highly lethal and highly virulent clades because clade I strains are generally considered the deadliest clade with less virulence, whereas clade II strains are highly virulent but less lethal [63,64,65]. Accordingly, the complete genome of Clade I Zaire-96-l-16 and seven variants of Clade II, including Clade IIa USA 2003, Clade llb-A Nigeria 2018, Clade IIb-A.2 USA 2022, Clade IIb-A.1 UK 2018 (UK P2), Clade IIb-A.1.1 USA 2021, and Clade IIb-B.1 France 2022, were employed in the IEDB Epitope Conservancy Tool. Almost all the epitopes selected for the vaccine construct exhibited 100% conservancy. Only one HTL epitope, SMVFEYRASTVIKGP, presented a single amino acid substituted from valine to isoleucine. The valine in the Clade I strains was substituted by isoleucine in Clade IIa, Clade llb-A, Clade IIb-A.2, Clade IIb-A.1, Clade IIb-A.1.1, and Clade IIb-B.1. However, valine-to-isoleucine substitution had a negligible effect on the protein because both valine and isoleucine are categorized as hydrophobic amino acids.

The vaccine construct model constructed from selected epitopes was improved further with adjuvants and linkers. Studies have shown that linkers incorporated into vaccine constructs can enhance vaccine stability [66,67]. These linkers are thought to function by facilitating spatial separation between individual epitopes, thereby minimizing potential steric hindrance and ensuring the independent immunogenicity of each epitope upon vaccination [66]. Furthermore, adjuvants were also fused to the vaccine model to improve the immune response. Moreover, adjuvants can exert immunostimulatory effects through various mechanisms, including the antigen depot-mediated promotion of prolonged exposure to the immune system, the activation of innate immunity via pathogen recognition receptor (PRR) engagement, immune cell costimulation, immunomodulation, and particularly, antigen-presenting cell (APC) maturation [68,69]. The results of the physicochemical property analyses in this study could indicate the improvement of the vaccine model, especially the best selected model with PADRE adjuvants. Compared to other designed vaccine models, the PADRE adjuvant vaccine had the highest antigenicity, highest solubility, and improved stability without being potentially allergenic or toxic.

After that, the PADRE adjuvant vaccine model was further subjected to molecular docking and MD with TLR4 because this TLR4 plays a crucial role in antiviral infection [70,71]. The TLR4 signaling complex might physically and hydrophobically interact with the hydrophobic pocket of a coreceptor called MD-2 and hydrophobic fusion peptides on viral proteins [70]. Another potential factor is glycosylation, as all TLR4-activating viral proteins undergo glycosylation [70]. Moreover, the way viral proteins assemble into units (oligomeric state) might also play a role in the interaction with the TLR4 signaling complex. [70]. Therefore, the TLR4 pdb file was retrieved from the Protein Data Bank (PDB ID: 4G8A); however, some missing residues were indicated by the pdb file information. Consequently, Modeler10.4 [72,73] and EMBOSS NEEDLE [74] were used to perform pdb file improvements before docking and MD.

The results of molecular docking with the ClusPro 2.0 server [75] demonstrated a favorable score of −1375.4 for both the center energy score and lowest energy. Similarly, the binding energy and dissociation constant determined by the HADDOCK 2.4 server [76,77,78] were favorable, with a binding energy (ΔG) of −18.7 kcal/mol and K_D_ of 2.00 × 10^−14^ M. Nevertheless, these docking servers have limitations. With rigid-body estimation, only moderate molecular conformations were allowed upon binding [79]. In fact, an important aspect of correctly approximating binding geometries is flexibility [80]. Additionally, a reliable approximation of key thermodynamic observations (such as the binding free energy) is not possible [80]. Furthermore, only a rigid picture of the binding process is produced, leading to a lack of approximation of kinetic quantities [80]. To address these limitations, the iMOD server [81] was used to analyze the rigidity and deformability of the vaccine-TLR4 complex, which indicated its stability. In addition, GROMACS software [82,83] was used to analyze the stability and dynamics of the complex structure within a timeframe of 50 ns and under more realistic conditions of solvation, neutralization (with Na and Cl ions), 310 K temperature, and 1 atm pressure. Moreover, immunization simulations performed with the C-ImmSim server [84,85,86] supported the idea that the PADRE vaccine has the potential to bind effectively to TLR4 because the results of immunization indicated both innate and adaptive immune responses. However, computational analysis is not sufficient to yield accurate results. In vitro characterization and in vivo studies are indispensable for validating the efficacy of broad-spectrum multi-epitope vaccine design.

The critical steps include protein expression in *E. coli*, to confirm its expression as predicted by its physicochemical properties, as well as in vivo studies in model organisms. Furthermore, the potential of this vaccine design can be significantly enhanced by exploring advanced delivery systems such as mRNAs and outer membrane vesicles (OMVs). These additional studies will not only affirm the efficacy and safety of the vaccine but also open new avenues for innovative approaches in vaccine development, offering a robust defense against diverse and evolving viral threats.

## 4. Materials and Methods

### 4.1. Retrieval of Monkeypox Viral Proteins

The protein sequences of the monkeypox virus Clade I (Congo/Central Africa) were retrieved from the GenBank database with the accession number DQ011154.1. This clade has the highest lethality and causes severe symptoms compared to other virulent clades. [63,64]. Although clade IIb caused the global pandemic in 2022, it had a lower mortality rate than clade I [63,64]. However, infected patients in clade I were recently observed in the Democratic Republic of the Congo between 2023 and 2024 [63,65,87]. According to the lethality rate and reemergence of Clade I, the MPXV strain Congo_2003_358 was utilized as a benchmark for the collection of MPX proteins.

Out of the 200 protein candidates, 24 proteins, including structural and nonstructural proteins retrieved from the complete genome, were selected based on antigenicity, allergenicity, and toxicity. The VaxiJen v2.0 [88,89,90] server (http://www.ddg-pharmfac.net/vaxijen/VaxiJen/VaxiJen.html) was used to screen for antigenic viral proteins. This model was evaluated using both internal leave-one-out cross-validation and external validation on test sets, achieving a predictive accuracy of up to 89% [89]. Additionally, AllergenFP v1.0 (https://ddg-pharmfac.net/AllergenFP/) was used to predict the allergenicity of proteins. This method was validated with known and unknown antigens, and the results demonstrated a prediction accuracy of 88% with a Matthews correlation coefficient of 0.759 [91] Only proteins with an antigenic probability score greater than 0.4 and no allergies were further utilized for epitope prediction.

### 4.2. Prediction of Cytotoxic T Lymphocytes (CTLs), Helper T Lymphocytes (HTLs), and Linear B Lymphocytes (LBLs)

The cytotoxic T-lymphocyte (CTL) epitopes of the selected MPXV proteins were predicted by using both NetMHCpan 4.1 BA and NetMHCpan 4.1 EL (https://services.healthtech.dtu.dk/services/NetMHCpan-4.1/) as prediction methods [92]. The HLA alleles, which included HLA-A0101, HLA-A0201, HLA-A0301, HLA-A2402, HLA-A2601, HLA-B0702, HLA-B0801, HLA-B2705, HLA-B3901, HLA-B4001, and HLA-B5801, were selected for screening of CTL epitopes. Since the vast majority of peptides presented by MHC class I include nine amino acids, the peptide length was set at nine amino acids. Additionally, only CTL epitopes demonstrating strong binding results were selected. The ability of the predicted epitopes with strong binding potential to trigger immunogenicity was further investigated via the Class-I Immunogenicity tool in the IEDB (http://tools.iedb.org/immunogenicity/) [93] before further assessment.

Similarly, the helper T-lymphocyte (HTL) epitopes of each viral protein were predicted by NetMHCpanII 4.1 BA and NetMHCpanII 4.1 EL (https://services.healthtech.dtu.dk/services/NetMHCIIpan-4.1/) [94]. DRB1_0101, DRB1_0301, DRB1_0401, DRB1_0701, DRB1_0801, DRB1_0901, DRB1_1001, DRB1_1101, DRB1_1201, DRB1_1301, DRB1_1501, and DRB1_1602 were selected as targeted HLA alleles to screen for HTL epitopes. The length of the epitopes was set to 15 amino acids by default. Only the epitopes with strong binding results were selected. The interferon-γ activation potential of the HTL epitopes was subsequently predicted by IFNepitope (http://crdd.osdd.net/raghava/ifnepitope/predict.php) [95].

Furthermore, the linear B-lymphocyte (LBL) epitopes of monkeypox proteins were predicted using the Bepipred Linear Epitope Prediction 2.0 server (https://services.healthtech.dtu.dk/services/BepiPred-2.0/) with the default threshold of 0.5 [96]. All the predicted epitopes (CTL, HTL, and LBL) were validated for allergenicity, antigenicity, and toxicity screening.

### 4.3. Effects of CTL, HTL, and B-Cell Epitopes on Allergenicity, Antigenicity, and Toxicity Parameters

The allergenicity of all the selected epitopes of HTL, CTL, and LBL was determined using AllergenFP v1.0 [91]. The VaxiJen v2.0 server was used to evaluate antigenicity with a threshold of 0.4 [88,89,90]. The ToxinPred2 server (https://webs.iiitd.edu.in/raghava/toxinpred2/) was used for toxicity prediction, with an SVM (Swiss-Prot)-based prediction method [97]. Only peptides demonstrating nonallergenic, antigenic, or nontoxic properties were selected for analysis of epitope conservation.

### 4.4. Screening of Transmembrane Regions

Before vaccine construction, candidate epitopes were predicted for transmembrane regions. The transmembrane regions of all the selected epitopes were predicted by DeepTMHMM 2.0 (https://dtu.biolib.com/DeepTMHMM) [37]. The predicted transmembrane segments were omitted from the vaccine construction to avoid low yields or unstable protein expression due to hydrophobic effects [98,99,100].

### 4.5. Epitope Conservation for the Broad-Spectrum Coverage of Vaccines Across Different Clades

To generate a broad-spectrum vaccine against various clades of the monkeypox virus, all the selected CTL, HTL, and LBL epitopes were investigated for epitope conservation across seven monkeypox virus clades, namely, clade I (Zaire-96-l-16), clade IIa (USA 2003), clade llb-A (Nigeria 2018), clade IIb-A.2 (USA 2022), clade IIb-A.1 (UK 2018 P2), clade IIb-A.1 (USA 2021), and clade IIb-B.1 (France 2022), using the IEDB Epitope Conservancy Tool (http://tools.iedb.org/conservancy/) [101]. Only highly conserved epitopes demonstrating percent identity higher than 90% were retrieved for vaccine construction.

### 4.6. Autoimmune Screening by Homology Analysis of Epitopes with Human Proteomes

To avoid the possibility of inducing autoimmune diseases or cross-reactivity in the host, all the selected CTL, HTL, and LBL epitopes obtained in Section 2.5 were blasted against the Homo sapiens proteome (Taxonomy ID: 9606) via the Peptide Search server (https://www.uniprot.org/peptide-search) [102]. Only epitopes exhibiting nonhomologous identity to the human proteome were subsequently selected for vaccine construction.

### 4.7. Population Coverage

To quantify the global impact of the vaccine, the IEDB population coverage tool (http://tools.iedb.org/tools/population/iedb_input) was used to analyze the worldwide distribution of HLA alleles targeted by our selected MHC class I and II epitopes [103]. The HLA allele genotypic frequencies from the Allele Frequency database (http://www.allelefrequencies.net/) were used to provide information on genetic variations in human populations [104]. This database offers data for more than 115 countries and categorizes 21 ethnicities within 16 geographic regions [104]. It also allows users to define custom populations with specific allele frequencies. The program calculates population coverage for various groups and generates an average value.

### 4.8. Construction of the Multi-Epitope Subunit Vaccine

The multi-epitope vaccines were constructed by combining the final epitopes of CTL, HTL, and LBL with separate linkers (AAY, HEYGAEALERAG, GPGPG, and KK). To enhance immunogenicity and prolong innate and adaptive immunity [39,40], a peptide adjuvant was fused to the multi-epitope vaccine. In this study, we selected five peptide adjuvants, including PADRE, beta-defensin 3, 50S ribosomal protein L7/12, RS-09, and the cholera toxin B subunit (CTB), to be incorporated N-terminally with our multi-epitope vaccine designs. Each adjuvant was N-terminally fused to a multi-epitope vaccine, yielding six multi-epitope vaccine constructs.

### 4.9. Physicochemical, Antigenicity, Allergenicity, Toxicity, and Solubility of the Vaccine

The physiochemical properties of each multi-epitope vaccine construct, including molecular weight, grand average hydropathicity (GRAVY), theoretical isoelectric point (pI), instability index, and half-life, were evaluated using the ProtParam tool of the ExPASy database server (https://web.ExPASy.org/protparam/) [49]. The antigenicity and toxicity of the designed vaccines were determined using AllergenFP v. 1.0 (http://ddg-pharmfac.net/AllergenFP/) [91] and the ToxinPred2 server (https://webs.iiitd.edu.in/raghava/toxinpred2/) [97], respectively. ToxinPred2 integrates three complementary methodologies, allowing it to attain a Matthews correlation coefficient exceeding 0.91 and achieve an area under the ROC curve close to 0.99 [105].

Allergenicity was identified by VaxiJen v2.0 (http://www.ddg-pharmfac.net/vaxijen/VaxiJen/VaxiJen.html) [88,89,90]. The SOLpro server (http://scratch.proteomics.ics.uci.edu/) was used to predict the solubility of the designed vaccines via an SVM-based method [35]. Additionally, the SignalP6.0 server (https://services.healthtech.dtu.dk/services/SignalP-6.0/) was used to identify signal peptide regions of the recombinant proteins [38].

### 4.10. Secondary Prediction, Tertiary Structure Predictions, Refinement, and Validation

The secondary structure features (α-helixes, β-sheets, and random coils) of the selected vaccine constructs were predicted using the PSIPRED server (http://bioinf.cs.ucl.ac.uk/psipred/) [106]. The tertiary structure of the selected vaccine was predicted using AlphaFold3 by implementing a fast homology search of MMseqs2 with AlphaFold3 [107,108]. The AlphaFold 3 improves protein–protein interaction predictions compared to the previous AF-Multimer, achieving 77% accuracy versus 67%, with particularly enhanced performance in protein–antibody predictions [108].

Following the initial three-dimensional (3D) modeling, the primary 3D models with the highest scores were further refined by submission to the GalaxyRefine module on the GalaxyWEB server (https://galaxy.seoklab.org/) [109,110]. This tool achieved the highest improvement in local structure quality based on the CASP10 evaluation. It enhances unreliable loops and terminal regions using an optimization-driven refinement approach and produces five refined models for each initial structure [109,111]. The model with the lowest MolProbity score was selected as the most well-defined model for each vaccine construct. These chosen models were further evaluated for quality validation using ERRAT and PROCHECK from the SAVES v6.0 (https://saves.mbi.ucla.edu/) and ProSA-Web (https://prosa.services.came.sbg.ac.at/prosa.php) datasets.

The ERRAT program evaluated overall structural integrity by quantifying nonbonded interactions between neighboring atoms [112]. Models with an overall quality factor exceeding 85 were considered reliable. The Ramachandran plot generated by the PROCHECK tool was used to evaluate the stereochemical accuracy of the protein backbone, with more residues located in favored regions indicating increased model accuracy [113,114,115,116]. Finally, through z-score prediction, the ProSA-Web server provided an overall quality assessment in which positive values suggested potential structural issues [117,118].

### 4.11. Molecular Docking of Toll-Like Receptor 4 (TLR4) and the Lead Vaccine Construct

To determine whether the lead vaccine can trigger an immune response via TLR4, molecular docking evaluations between the lead vaccine and TLR4 (PDB ID: 4G8A) were performed to predict the occurrence of persistent interactions between the complexes [119].

To bridge the gaps caused by missing residues in the retrieved PDB file of TLR4, MODELLER10.4 was used to locate missing amino acid residues prior to molecular docking analysis to obtain the most accurate outcomes [72,73]. After the PDB file was improved, the ClusPro 2.0 server (https://cluspro.org/login.php) was used to measure the binding energy [75] between the lead multi-epitope vaccine and the TLR4 receptor. Thirty model structures were generated for the docking scenario, and the ten models demonstrating the largest clusters were prioritized for further analysis.

To enhance structural precision, the top-ranked model obtained from the ClusPro 2.0 server was further refined using the HADDOCK 2.4 server (https://wenmr.science.uu.nl/haddock2.4/) [76,120]. This platform integrates diverse experimental and computational data to reliably generate high-quality models of macromolecular complexes [120]. It further optimizes these structures by systematically rearranging side chains within defined interfaces and employing soft rigid-body refinement techniques to enhance structural accuracy [76,120]. Subsequently, the binding energies of the refined docked models were assessed using the PRODIGY server (https://bianca.science.uu.nl/prodigy/), which provided quantitative insights into the binding affinities of the peptides [76,77,78]. PyMOL was used to visualize the structures of the figures.

### 4.12. In Silico Immune Stimulation

The selected vaccine was analyzed via the C-IMMSIM tool (https://kraken.iac.rm.cnr.it/C-IMMSIM/) to predict the immune efficacy of our designed vaccine [84,85,86]. The C-IMMSIM server can identify epitopes through position-specific scoring matrix (PSSM) approaches to simulate immune interactions [84,85,86]. For the immune simulation, the simulation time steps were set to 1050, where the 1-time step equals 8 h. The time step of each injection point was set at 1, 84, or 168. The dose of each antigen injection was set at 1000, and the time interval between each injection was four weeks, which is the recommended interval for vaccination.

### 4.13. Codon Optimization and In Silico Cloning

The vaccine construct was subjected to codon optimization by using the Java Codon Adaptation Tool (JCAT) (http://jcat.de) [82] with *E. coli* strain K12 as the organism. After codon optimization, the vaccine sequence was inserted into the expression vector pET21a (+), which was digested with NdeI and XhoI using Snap Gene 7.2.1.

### 4.14. Molecular Dynamics (MD) Simulation

The best docking model obtained from molecular docking analysis was analyzed by the iMODS server to predict the stability of the vaccine-TLR complex based on normal mode analysis (NMA) to predict rigidity and deformability [81]. The iMODs server provides information on deformability graph, B-factor graph (RMSD values), eigenvalues, variance plot, covariance map, and elastic network map. Additionally, the structural dynamics and stability of the docked vaccine-TLR complex were analyzed by using GROMACS v2024 software [82,83]. The CHARMM36 forcefield [121,122] was adopted to create the topology file for MD analysis. The TIP3P water model [122,123,124] was also utilized for solvation of the vaccine-TLR complex. Afterwards, Na and Cl ions were added to neutralize the system, followed by minimizing the system energy. The NVT of each system was performed at 310 K, and the modified Berendsen thermostat [125] was used to perform temperature equilibration. A Parrinello–Rahman barostat [126] was also employed. The pressure was adjusted to 1 atm for the NPT simulation with a time constant of 2 ps. Then, the Leapfrog algorithm was used to integrate the equations of motion with a time step of 2.0 fs.

The vaccine-TLR complex system was run for a 50 ns simulation under constant temperature and pressure. The LINCS algorithm [127] was used for the constraint of hydrogen bonds. The particle mesh Ewald (PME) method [128] was used to analyze the long-range electrostatic energy. Additionally, short-range Coulomb electrostatic and van der Waals cutoffs of 1.0 nm were used to perform MD. The results of the MD analysis performed with GROMACS software included the root mean square deviation (RMSD), root mean square fluctuations (RMSF), radius of gyration (Rg), hydrogen bonds, and total solvent accessible surface area (SASA) of the system.

## Figures and Tables

**Figure 1 ijms-26-07210-f001:**
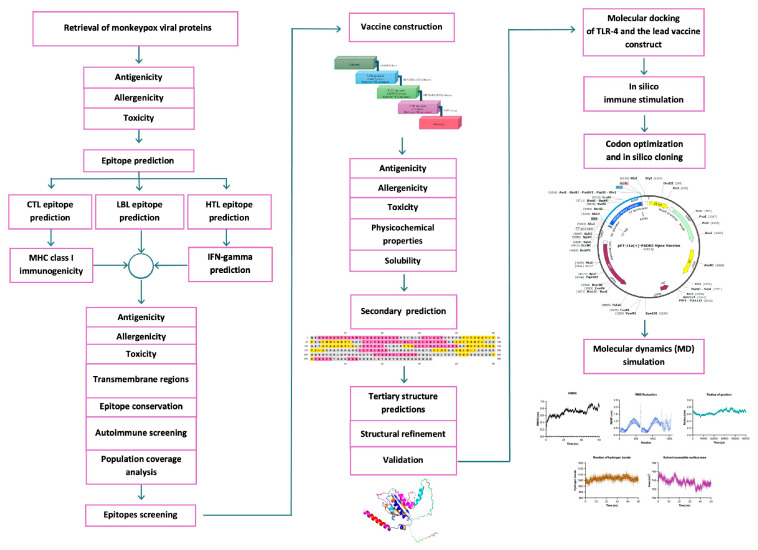
Schematic workflow for designing a broad-spectrum, multi-epitope pan-Mpox vaccine.

**Figure 2 ijms-26-07210-f002:**
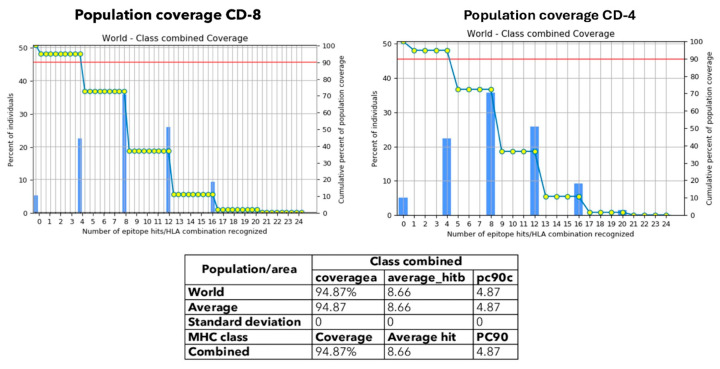
Worldwide population coverage of selected CTL and HTL epitopes for vaccine development. Population coverage analysis of the selected CTL and HTL epitopes was performed using the IEDB tool, based on the distribution of HLA class I and II alleles.

**Figure 3 ijms-26-07210-f003:**
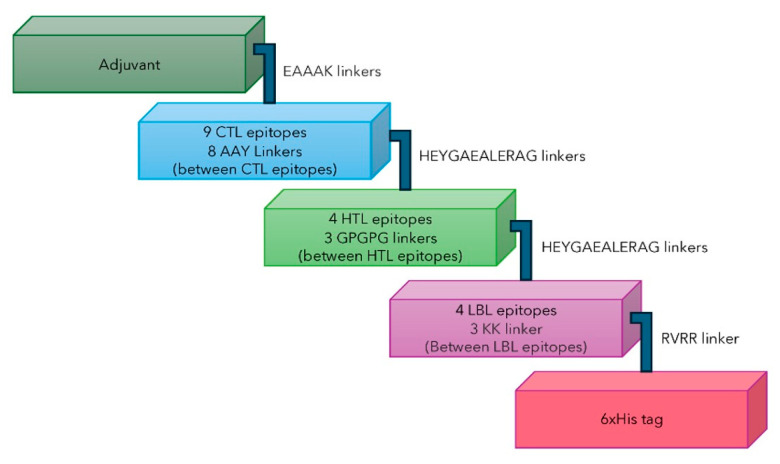
Schematic illustration of the multi-epitope MPXV vaccine constructs consisting of adjuvant, CTL, HTL, and LBL epitopes; a 6xHis tag; and linkers. Adjuvant, CTL, HTL, LBL, and the 6xHis tag are represented by dark green, light blue, green, purple, and pink, respectively.

**Figure 4 ijms-26-07210-f004:**
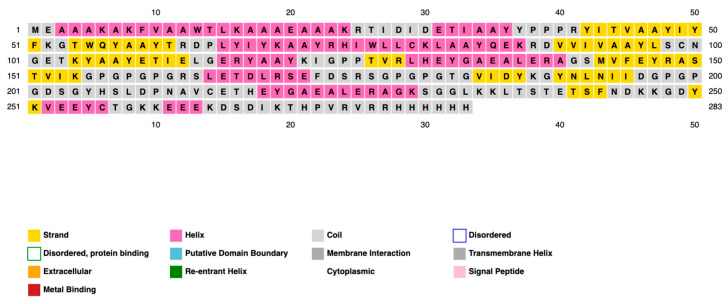
Secondary structures of the designed PADRE vaccine analyzed with the PSIPRED 4.0 server.

**Figure 5 ijms-26-07210-f005:**
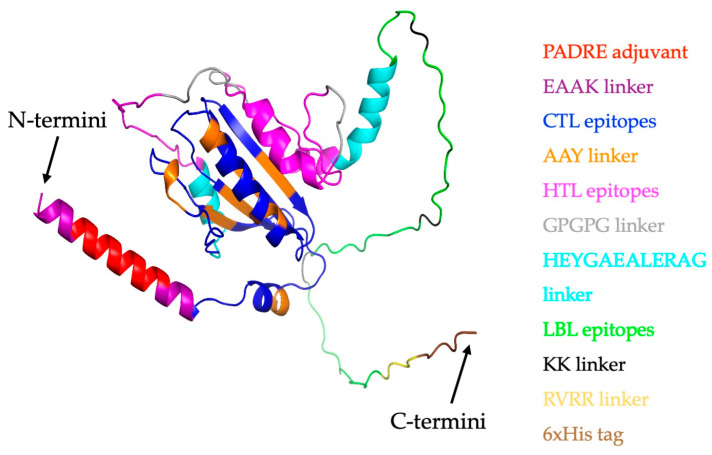
The final 3D structure of the PADRE-Mpox vaccine, predicted using AlphaFold 3 and subsequently refined with the GalaxyWEB server, is illustrated. In the visualization, different components are color-coded: PADRE adjuvant (red), EAAK linker (purple), CTL epitopes (blue), AAY linker (orange), HTL epitopes (magenta), GPGPG linker (grey), HEYGAEALERAG linker (cyan), LBL epitopes (green), KK linker (black), RVRR linker (yellow), and the 6xHis tag (brown).

**Figure 6 ijms-26-07210-f006:**
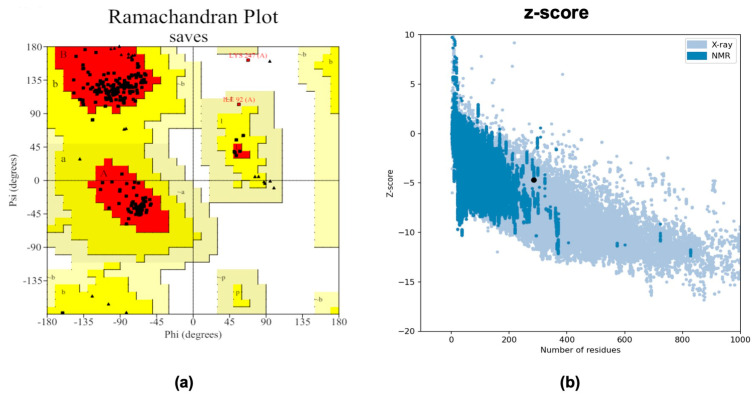
Validation of the lead-refined model with the Ramachandran plot and z-score. (**a**) The Ramachandran plot of the selected refined model representing favored regions, allowed regions, generously allowed regions and disallowed regions. (**b**) Z-score generated by the ProSa-web server.

**Figure 7 ijms-26-07210-f007:**
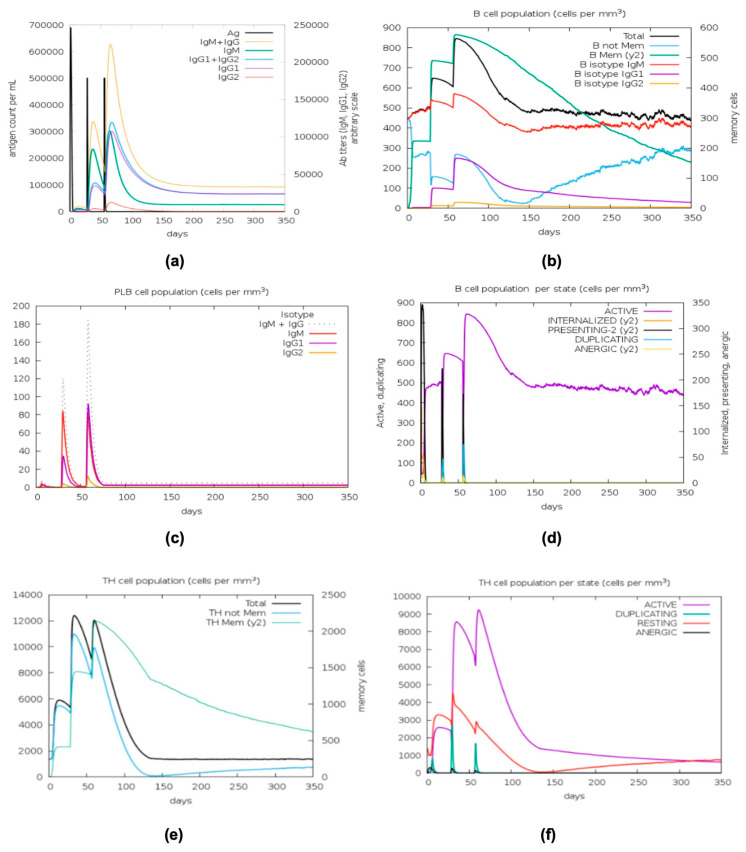
The graphs representing the immunization simulation by the C-IMMSIM server. (**a**–**d**) Levels of the B lymphocyte population after vaccination. (**e**,**f**) Helper T-cell population after vaccination. (**g**) Cytotoxic T-cell population after vaccination. (**h**) The levels of total NK cells, (**i**) total MA population per state, and (**j**) DC population per state. (**k**) Levels of cytokines and interleukin concentrations, with the inset plot indicating the danger signal together with leukocyte growth.

**Figure 8 ijms-26-07210-f008:**
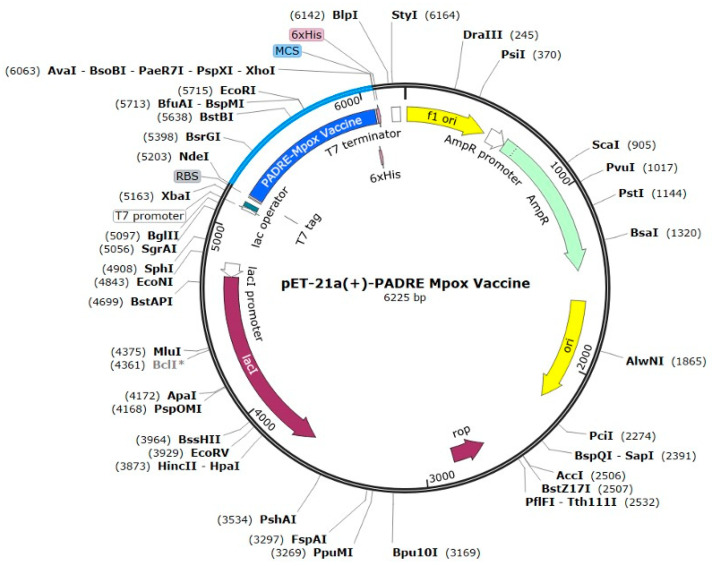
In silico cloning map showing the insertion of the vaccine sequence (shown in blue) into the pET21a (+) expression vector.

**Figure 9 ijms-26-07210-f009:**
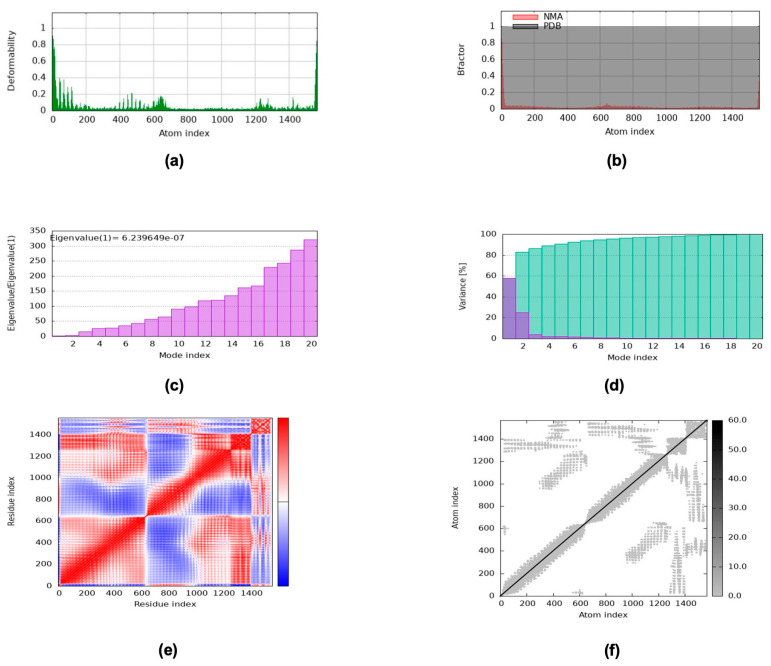
Graphs representing normal mode analysis (NMA) of the iMOD server. (**a**) The main-chain deformability. (**b**) B-factor graph (RMSD values). (**c**) Eigenvalue for structural deformation. (**d**) Individual variance and cumulative variance. (**e**) Atomic motion in the regions of MD. (**f**) Maps of the elastic network.

**Figure 10 ijms-26-07210-f010:**
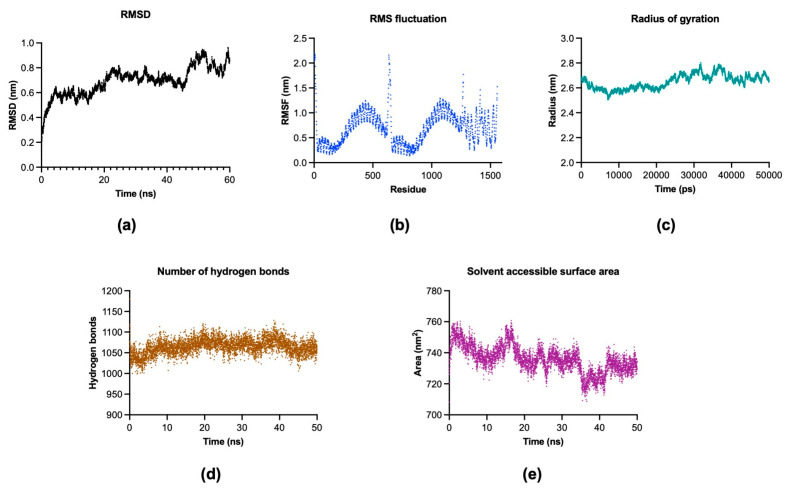
Graphs illustrating the results of molecular dynamics using GROMACS software. (**a**) RMSD, (**b**) RMSF, (**c**) Rg, (**d**) number of H bonds, and (**e**) the SASA.

**Table 1 ijms-26-07210-t001:** Selected monkeypox proteins based on antigenic and nonallergenic properties.

**Structural Protein**			
**Protein Name**	**Product**	**Antigenicity**	**Allergenicity**
A14L	IMV membrane protein	0.5019	Non-allergen
A15L	Phosphorylated IMV membrane protein	0.4759	Non-allergen
A15.5L	IV and IMV membrane protein	0.7480	Non-allergen
A30L	IMV surface protein (envelope protein)	0.6212	Non-allergen
A40L	CD47-like putative membrane protein	0.4361	Non-allergen
B6R	EEV type-I membrane glycoprotein	0.5786	Non-allergen
B16R	IFN-alpha/beta-receptor-like secreted glycoprotein	0.5407	Non-allergen
B19R	Serine protease inhibitor-like SPI-1 protein	0.7784	Non-allergen
B21R	Putative membrane-associated glycoprotein	0.5246	Non-allergen
C2L	Serine protease inhibitor-like protein SPI-3	0.5702	Non-allergen
F7R	Membrane protein	0.4334	Non-allergen
H3L	IMV heparin binding surface protein	0.4538	Non-allergen
L5L	Llate 16 kDa putative membrane protein	0.7559	Non-allergen
M1R	Myristyl protein	0.6339	Non-allergen
M5R	Putative membrane protein	0.4468	Non-allergen
**Nonstructural Protein**			
**Protein Name**	**Product**	**Antigenicity**	**Allergenicity**
A28L	A-type inclusion protein	0.4602	Non-allergen
D3R	Secreted epidermal growth factor-like protein	0.4947	Non-allergen
D10L	Host-range	0.5965	Non-allergen
E4R	Uracil-DNA glycosylase	0.5759	Non-allergen
F1L	Poly-A polymerase catalytic subunit VP55	0.4419	Non-allergen
F3L	Double-stranded RNA binding protein	0.4127	Non-allergen
F7R	Membrane protein	0.4334	Non-allergen
F8L	DNA polymerase	0.4103	Non-allergen
J3R/J1L	Chemokine-binding protein	0.7563	Non-allergen

**Table 2 ijms-26-07210-t002:** (**a**) Finalized CTL epitopes for multi-epitope vaccine construction. (**b**) Finalized HTL epitopes for multi-epitope vaccine construction. (**c**) Finalized LBL epitopes for multi-epitope vaccine construction.

(**a**)							
**Protein**	**CTL Epitope**	**Antigenic Score**	**Allergenicity**	**Toxicity**	**Class I Immunogenicity Score**	**Binding Affinity**	**Human Proteome**
A40L	RHIWLLCKL	0.5947	NA	NT	0.0734	Strong Binding	NH
B6R	LSCNGETKY	0.8019	NA	NT	0.0100	Strong Binding	NH
B9R	KIGPPTVRL	1.1705	NA	NT	0.0907	Strong Binding	NH
C2L	IYFKGTWQY	0.8395	NA	NT	0.0098	Strong Binding	NH
	TRDPLYIYK	0.4908	NA	NT	0.0919	Strong Binding	NH
F8L	RTIDIDETI	1.0682	NA	NT	0.3232	Strong Binding	NH
	YPPPRYITV	0.4807	NA	NT	0.1672	Strong Binding	NH
	ETIELGERY	1.4370	NA	NT	0.2798	Strong Binding	NH
H3L	QEKRDVVIV	1.4396	NA	NT	0.1551	Strong Binding	NH
(**b**)							
**Protein**	**HTL Epitope**	**Antigenicity**	**Allergenicity**	**Toxicity**	**IFN-γ Epitope**	**Binding Affinity**	**Human Proteome**
B6R	DSGYHSLDPNAVCET	0.9057	NA	NT	Positive	Strong Binding	NH
E4R	TGVIDYKGYNLNIID	1.4293	NA	NT	Positive	Strong Binding	NH
F8L	SMVFEYRASTVIKGP	0.5161	NA	NT	Positive	Strong Binding	NH
	RSLETDLRSEFDSRS	0.9620	NA	NT	Positive	Strong Binding	NH
(**c**)							
**Protein**	**LBL Epitope**	**Antigenicity**		**Allergenicity**		**Toxicity**	**Human Proteome**
B6R	LTSTETSFND	1.5197		NA		NT	NH
B9R	GDYKVEEYCTG	0.9854		NA		NT	NH
H3L J3R/J1L	KSGGL	2.1517		NA		NT	NH
	EEEKDSDIKTHPV	0.6156		NA		NT	NH

NA = non-allergen, NT = non-toxin, NH = non-homologous.

**Table 3 ijms-26-07210-t003:** Antigenicity, allergenicity, toxicity, and physicochemical properties of the vaccine candidates.

Properties	Adjuvant				
	Cholera Toxin B Subunit (CTB)	PADRE	RS09	50S Ribosomal L7/L12	Beta-Defensin
Antigenicity (VaxiJen 2.0)	0.5979	0.6419	0.6340	0.5566	0.6321
Allergenicity (AllergenFP v.1.0)	Non-allergen	Non-allergen	Non-allergen	Non-allergen	Non-allergen
Toxicity (ToxinPred V.2)	Non-Toxin	Non-Toxin	Non-Toxin	Non-Toxin	Toxic
Number of amino acids	373	283	277	400	330
Molecular weight (Da)	41,756.08	31,458.34	30,802.50	43,551.20	37,049.07
Theoretical isoelectric point (pI)	6.92	7.17	6.64	5.60	8.83
The estimated half-life1	30 h	30 h	30 h	30 h	30 h
The estimated half-life2	>20 h	>20 h	>20 h	>20 h	>20 h
The estimated half-life3	>10 h	>10 h	>10 h	>10 h	>10 h
Aliphatic index	72.57	68.76	67.76	79.17	67.18
Instability index	34.40	30.64	34.02	29.25	36.61
Grand average of hydropathicity (GRAVY)	−0.566	−0.586	−0.642	−0.387	−0.632
Solubility	0.83	0.93	0.87	0.98	0.72
Signal Peptide	No signal peptide	No signal peptide	No signal peptide	No signal peptide	No signal peptide

The estimated half-life1 = mammalian reticulocytes, in vitro, The estimated half-life2 = yeast, in vivo, The estimated half-life3 = Escherichia coli, in vivo.

**Table 4 ijms-26-07210-t004:** The docked PADRE vaccine-TLR4 complex was analyzed by ClusPro 2.0, and refined docking analysis was performed with the HADDOCK 2.4 server.

Tool	Parameter	PADRE Vaccine-TLR4 Complex
ClusPro 2.0	Center	−1375.4
	Lowest Energy	−1375.4
HADDOCK 2.4 server	K_D_ (M) at 25 °C	2.00 × 10^−14^
	ΔG (kcal mol^−1^)	−18.7
	HADDOCK score	−808.4 ± 7.9
	Cluster size	20
	RMSD from the overall lowest-energy structure	0.7 ± 0.4
	Van der Waals energy	−426.0 ± 3.7
	Electrostatic energy	−1417.1 ± 28.0
	Desolvation energy	−99.1 ± 6.6
	Restraints violation energy	0.0 ± 0.0
	Buried surface area	12,977.8 ± 131.7
	Z-score	0

## Data Availability

All data generated or analyzed in this study are provided within the manuscript.

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
