# Peer review of "Development of a Broad-Spectrum Pan-Mpox Vaccine via Immunoinformatic Approaches"

_ijms, 2025, doi:10.3390/ijms26157210_

Round 1
Reviewer 1 Report
Comments and Suggestions for Authors
Reviewer Report on “Development of a broad-spectrum pan-Mpox vaccine via immunoinformatic approaches” by Puagsopa et al.
The study has the potential to serve a broad scientific readership, but several issues must be addressed to improve its quality and rigor.
The authors used VaxiJen 2.0 and AllergenFP 1.0 to select candidate sequences. Because these tools underpin all subsequent analyses, their predictive performance should be validated. Please benchmark their results against relevant experimental data from the literature or provide a brief discussion (supplementary material is acceptable) that justifies their reliability for this application.
Predictions were made with DeepTMHMM and SignalP 6.0. Include a short evaluation of these tools’ accuracy on comparable datasets and, where possible, cross-check your results with published experimental findings.
Secondary-structure prediction: This section can be streamlined. Since the 3-D models already reveal the secondary-structure elements, a concise summary will suffice; lengthy narrative descriptions are unnecessary.
Merge Tables 2a, 2b, and 2c into a single table with clear column headings to eliminate redundancy.
Replace the shorthand “Rama” with the full term “Ramachandran plot.”
Model refinement. Provide a more rigorous assessment for e.g., include molecular-dynamics (MD) simulations or additional quality metrics. Revise Figure 4 to show a before-and-after comparison with a clear superposition that highlights structural improvements not just the structure alone.
Revise the sentence “The ClusPro 2.0 server was used to predict the binding affinity between the PADRE-302 vaccine and TLR4.” ClusPro produces docking poses and scores, not absolute binding affinities. Specify the metric you report (e.g., ClusPro weighting score or buried surface area) and avoid labeling it “binding affinity” unless further validated.
While the diverse tool set demonstrates thoroughness, each server’s performance should be substantiated. Provide benchmarks or cite validation studies that show how well these tools perform on problems similar to yours.
Results and Discussion: The current narrative is overly broad. Focusing on the most critical findings will yield a sharper, more coherent discussion aligned with your study’s aims.
Addressing these points will substantially enhance the reliability and impact of the manuscript.
Author Response
RESPONSE TO REVIEWER
We appreciate the thorough evaluation of our manuscript by the editor and reviewers and their overall assessment of our work. In addition, text in introduction, results, discussion, materials and methods sections, and figure caption have been modified based on reviewer feedback. We hope the editor and reviewers will agree that our manuscript is greatly improved thanks to these revisions. A point-by-point response to the reviewer comments is provided below, and changes made to the manuscript text are marked in blue (in the attached) and in the accompanying manuscript file.
- The authors used VaxiJen 2.0 and AllergenFP 1.0 to select candidate sequences. Because these tools underpin all subsequent analyses, their predictive performance should be validated. Please benchmark their results against relevant experimental data from the literature or provide a brief discussion (supplementary material is acceptable) that justifies their reliability for this application.
Ans. We appreciate the reviewer raising this point. We have included a brief discussion that justifies the reliability of VaxiJen 2.0 and AllergenFP 1.0.
Page 24 Line 633-640
The VaxiJen v2.0 [92–94] server (http://www.ddg-pharmfac.net/vaxijen/VaxiJen/VaxiJen.html) was used to screen for antigenic viral proteins. This model was evaluated using both internal leave-one-out cross-validation and external validation on test sets, achieving a predictive accuracy of up to 89% [93]. Additionally, AllergenFP v1.0 (https://ddg-pharmfac.net/AllergenFP/) was used to predict the allergenicity of proteins. This method was validated with known and unknown antigen, the result demonstrated the prediction accuracy of 88% with the Matthews correlation coefficient of 0.759 [95].
- Predictions were made with DeepTMHMM and SignalP 6.0. Include a short evaluation of these tools’ accuracy on comparable datasets and, where possible, cross-check your results with published experimental findings.
Ans. We thank the reviewer for this feedback.
Page 5 Line 177-182
The DeepTMHMM 2.0 platform has demonstrated superior performance compared to other methods across most protein types [36]. Furthermore, SignalP 6.0 was evaluated in comparison with existing prediction tools, which, in contrast to SignalP 6.0, lack the capability to identify all classes of signal proteins. Additionally, the outcomes from these tools exhibited inherent biases, as they could not be assessed within a cross-validation framework [37].
- Secondary-structure prediction: This section can be streamlined. Since the 3-D models already reveal the secondary-structure elements, a concise summary will suffice; lengthy narrative descriptions are unnecessary.
Ans. We thank the reviewer for these helpful suggestions. We have condensed the section on secondary-structure prediction in accordance with the reviewer’s recommendation.
Page 10 Line 299-300
The PSIPRED 4.0 server predicted our vaccine construct to include 44.6% coils, 33.7% α-helices, and 21.8% β-strands, as shown in Fig. 4.
- Merge Tables 2a, 2b, and 2c into a single table with clear column headings to eliminate redundancy.
Ans. We thank reviewer for the feedback. After carefully pondering, it is hard to merge all three in one table because the information in each table is related to either CTL/HTL/LBL, which includes different information for their selected epitopes. Thus, we elected to maintain the existing Tables 2a, 2b, and 2c.
- Replace the shorthand “Rama” with the full term “Ramachandran plot.”
Ans. We thank reviewer for noting this point. We have corrected Rama to the full term Ramachandran plot in the manuscript.
- Model refinement. Provide a more rigorous assessment for e.g., include molecular-dynamics (MD) simulations or additional quality metrics. Revise Figure 4 to show a before-and-after comparison with a clear superposition that highlights structural improvements not just the structure alone.
Ans. We thank the reviewer for noting this important point.
Regarding model refinement, the 3D structure obtained through model refinement demonstrated significant improvement in terms of Molprobity and Ramachandran plot favored scores.
Page 10 Line 305-307
Model 1 was selected based on improved Molprobity and Ramachandran plot favored scores of 1.302 and 98.2, respectively, which indicated the high quality of the model.
Regarding Figure 4, the main secondary structures—such as the α-helix and β-sheet—remain largely unchanged. Only minor adjustments in the orientation of the random coil regions were made. To enhance clarity, we elected to update Figure 4 to display the refined 3D structure with color coding to distinguish the different structural components, as shown below.
Page 16 Line 453-458
Figure 5. The final 3D structure of the PADRE-Mpox vaccine, predicted using AlphaFold 3 and subsequently refined with the GalaxyWEB server, is illustrated. In the visualization, different components are color-coded: PADRE adjuvant (red), EAAK linker (purple), CTL epitopes (blue), AAY linker (orange), HTL epitopes (magenta), GPGPG linker (grey), HEYGAEALERAG linker (cyan), LBL epitopes (green), KK linker (black), RVRR linker (yellow), and the 6xHis tag (brown).
- Revise the sentence “The ClusPro 2.0 server was used to predict the binding affinity between the PADRE-302 vaccine and TLR4.” ClusPro produces docking poses and scores, not absolute binding affinities. Specify the metric you report (e.g., ClusPro weighting score or buried surface area) and avoid labeling it “binding affinity” unless further validated.
Ans. We thank the reviewer for these helpful suggestions. We have corrected the sentence according to the reviewer’s suggestion.
Page 11 Line 321
The ClusPro 2.0 server was used to provide the protein-protein docking.
- While the diverse tool set demonstrates thoroughness, each server’s performance should be substantiated. Provide benchmarks or cite validation studies that show how well these tools perform on problems similar to yours.
Ans. We thank the reviewer for this feedback, and we have included server’s performance, including the VaxiJen, AllergenFP v1.0, the DeepTMHMM 2.0, and SignalP 6.0 in the manuscript.
Page 24 Line 633-640
The VaxiJen v2.0 [92–94] server (http://www.ddg-pharmfac.net/vaxijen/VaxiJen/VaxiJen.html) was used to screen for antigenic viral proteins. This model was evaluated using both internal leave-one-out cross-validation and external validation on test sets, achieving a predictive accuracy of up to 89% [93]. Additionally, AllergenFP v1.0 (https://ddg-pharmfac.net/AllergenFP/) was used to predict the allergenicity of proteins. This method was validated with known and unknown antigen, the result demonstrated the prediction accuracy of 88% with the Matthews correlation coefficient of 0.759 [95].
Page 5 Line 177-182
The DeepTMHMM 2.0 platform has demonstrated superior performance compared to other methods across most protein types [36]. Furthermore, SignalP 6.0 was evaluated in comparison with existing prediction tools, which, in contrast to SignalP 6.0, lack the capability to identify all classes of signal proteins. Additionally, the outcomes from these tools exhibited inherent biases, as they could not be assessed within a cross-validation framework [37].
Page 26 Line 725-727
ToxinPred2 integrates three complementary methodologies, allowing it to attain a Matthews correlation coefficient exceeding 0.91 and achieve an area under the ROC curve close to 0.99 [109]
Page 27 Line 737-741
The tertiary structure of the selected vaccine was predicted using AlphaFold3 by implementing a fast homology search of MMseqs2 with AlphaFold3 [111,112]. The AlphaFold 3 improves protein-protein interaction predictions compared to the previous AF-Multimer, achieving 77% accuracy versus 67%, with particularly enhanced performance in protein-antibody predictions [112].
Page 27 Line 742-747
Following the initial three-dimensional (3D) modeling, the primary 3D models with the highest scores were further refined by submission to the GalaxyRefine module on the GalaxyWEB server (https://galaxy.seoklab.org/) [113,114]. This tool achieved the highest improvement in local structure quality based on the CASP10 evaluation. It enhances unreliable loops and terminal regions using an optimization-driven refinement approach and produces five refined models for each initial structure [113,115].
Page 28 Line 773-779
To enhance structural precision, the top-ranked model obtained from the ClusPro 2.0 server was further refined using the HADDOCK 2.4 server (https://wenmr.science.uu.nl/haddock2.4/) [80,124]. This platform integrates diverse experimental and computational data to reliably generate high-quality models of macromolecular complexes [124]. It further optimizes these structures by systematically rearranging side chains within defined interfaces and employing soft rigid-body refinement techniques to enhance structural accuracy [80,124].
- Results and Discussion: The current narrative is overly broad. Focusing on the most critical findings will yield a sharper, more coherent discussion aligned with your study’s aims.
Ans. We thank the reviewer for this valuable feedback. In response, we have revised the Results and Discussion section to focus more closely on the key findings. The narrative has been streamlined to enhance clarity and better align with the objectives of the study.
Reviewer 2 Report
Comments and Suggestions for Authors
The work of Bunyarit Meksiriporn’s lab uses immunoinformatics methods to develop a multi-epitope vaccine against Monkeypox virus (MPXV). The manuscript describes the steps of epitope selection, vaccine construction, molecular docking analyses, and molecular dynamics simulations. This includes validation using AlphaFold, ClusPro, and GROMACS.
I consider the work relevant but there are issues in the manuscript that I believe need to be improved before it is published.
1) In both the Introduction and the Discussion, it is necessary to include more information about MPXV and vaccines to give the reader an understanding of the context of this disease and what information is available. Discussing the proteins involved in immunogenicity and the regions studied seems to be important in this context.
Here are some examples:
Wang Y, Yang K, Zhou H. Immunogenic proteins and potential delivery platforms for mpox virus vaccine development: A rapid review. Int J Biol Macromol. 2023 Aug 1;245:125515. doi: 10.1016/j.ijbiomac.2023.125515. Epub 2023 Jun 21. PMID: 37353117; PMCID: PMC10284459.
Garcia-Atutxa I, Mondragon-Teran P, Huerta-Saquero A, Villanueva-Flores F. Advancements in monkeypox vaccines development: a critical review of emerging technologies. Front Immunol. 2024 Oct 11;15:1456060. doi: 10.3389/fimmu.2024.1456060. PMID: 39464881; PMCID: PMC11502315.
Zhang X, Liu DA, Qiu Y, Hu R, Chen S, Xu Y, Chen K, Yuan J, Li X. Recent Advances in Mpox Epidemic: Global Features and Vaccine Prevention Research. Vaccines (Basel). 2025 Apr 25;13(5):466. doi: 10.3390/vaccines13050466. PMID: 40432078; PMCID: PMC12116011.
(2) The text construction throughout the results is extremely long. Dividing them into smaller parts could make it easier to read.
(3) Epidemiological data needs to be updated. The manuscript cites 41,664 confirmed cases and five deaths, where the authors mention information up to July 2024. Including recent data could show the current epidemiological parameter.
(4) Although the epitope selection is well explained, it would be useful to include a flowchart summarizing the filtering process (antigenicity, allergenicity, toxicity, etc.). In addition, the rationale for the choice of adjuvants (PADRE, beta-defensin, etc.) could be expanded, highlighting why PADRE was selected as the best candidate.
(5) Overall, the figures (such as population coverage and molecular docking) are essential, but some captions are too brief. Including more detailed descriptions could help the reader interpret the results without having to search for information in the running text.
(6) Table A1 (in the Appendix) is extensive and difficult to follow. A simplified version or a summary of the most relevant epitopes would be desirable.
(7) Comparing the results with other recent studies of multi-epitope vaccines for MPXV would enrich the discussion. Please highlight the limitations of the study (such as the reliance on computational predictions and the need for experimental validation) as future research perspectives.
(8) Please review the grammar and names throughout the manuscript. Check for typos (e.g., "Vaxijen" instead of "VaxiJen" in some places).
(9) Please include a paragraph in the discussion about the general limitations of the methods used.
(10) The authors should review the molecular dynamics simulation part. I consider a 50 ns simulation too short for a conclusion of this size to be published. The authors should extend this simulation to 200 ns and check for stochastic convergence, mainly due to the fact that it has predictions of structures without real experimental data. The authors should include the radius of gyration (Rg) and check that Rg does not change abruptly over time.
Comments on the Quality of English Language
Review the grammar and names along the manuscript.
Author Response
RESPONSE TO REVIEWER 2
We appreciate the thorough evaluation of our manuscript by the editor and reviewers and their overall assessment of our work. In addition, text in introduction, results, discussion, materials and methods sections, and figure caption have been modified based on reviewer feedback. We hope the editor and reviewers will agree that our manuscript is greatly improved thanks to these revisions. A point-by-point response to the reviewer comments is provided below, and changes made to the manuscript text are marked in blue here and in the attached and in the accompanying manuscript file.
1) In both the Introduction and the Discussion, it is necessary to include more information about MPXV and vaccines to give the reader an understanding of the context of this disease and what information is available. Discussing the proteins involved in immunogenicity and the regions studied seems to be important in this context.
Here are some examples:
Wang Y, Yang K, Zhou H. Immunogenic proteins and potential delivery platforms for mpox virus vaccine development: A rapid review. Int J Biol Macromol. 2023 Aug 1;245:125515. doi: 10.1016/j.ijbiomac.2023.125515. Epub 2023 Jun 21. PMID: 37353117; PMCID: PMC10284459.
Garcia-Atutxa I, Mondragon-Teran P, Huerta-Saquero A, Villanueva-Flores F. Advancements in monkeypox vaccines development: a critical review of emerging technologies. Front Immunol. 2024 Oct 11;15:1456060. doi: 10.3389/fimmu.2024.1456060. PMID: 39464881; PMCID: PMC11502315.
Zhang X, Liu DA, Qiu Y, Hu R, Chen S, Xu Y, Chen K, Yuan J, Li X. Recent Advances in Mpox Epidemic: Global Features and Vaccine Prevention Research. Vaccines (Basel). 2025 Apr 25;13(5):466. doi: 10.3390/vaccines13050466. PMID: 40432078; PMCID: PMC12116011.
Ans. We thank the reviewer for this helpful input and for the insightful references, which have been incorporated into the manuscript. In response, we have added more detailed information regarding MPXV, relevant vaccines, and the proteins involved in immunogenicity.
Page 2 Line 106-113
Several MPXV proteins, including M1R (myristylprotein), B6R (EEV type-I membrane glycoprotein), and H3L (IMV heparin binding surface protein), are associated with immune responses within poxviruses and are considered attractive targets for vaccine development [23]. Additionally, B9R has been implicated in the pathogenesis of MPXV [24], whereas F8L (DNA polymerase) has been reported to stimulate the CD8+ T-cell population and the release of interferon-g in rhesus macaques [25]. Notably, a previous study demonstrated the use of M1R and B6R as components of a recombinant protein vaccine, and the results showed significant inhibition of MPXV in mice [26].
Page 21-22 Line 493-515
The Advisory Committee on Immunization Practices (ACIP) recommends the ACAM2000 live-attenuated vaccinia virus vaccine for specific personnel groups. These groups include laboratory personnel routinely handling human infectious orthopox viruses (monkeypox, cowpox, and variola) and medical personnel caring for vaccinia virus-infected patients [54]. However, ACAM2000 carries potential safety risks. Studies have reported occurrences of postvaccinal encephalitis, eczema vaccinatum, and progressive vaccinia following vaccination [54]. Additionally, the presence of live vaccinia virus raises concerns about secondary transmission to both vaccinated individuals and those in close contact and the potential for the development of myocarditis or pericarditis in some recipients [55]. Moreover, administration of the JYNNEOS vaccine has been associated with various side effects, including induration, chills, headache, sore throat, nausea, redness, myalgia, firmness/tightening, and pain [52]. A retrospective study conducted in a northwestern US population cohort (Oregon) identified 10 cases of cardiac events following JYNNEOS vaccination between July and October 2022 [56]. Similarly to JYNNEOS, LC16 is a live-attenuated, nonreplicating vaccine approved for monkeypox prevention in Japan in August 2022 [55]. Most adverse events were reported as mild to moderate localized or systemic in nature. Notably, clinical trials and cohort studies have not identified serious adverse events such as encephalitis or symptomatic myocarditis [52]. While preclinical data suggest potential safety for immunocompromised patients and atopic dermatitis patients, widespread vaccination in these populations is not currently recommended due to insufficient clinical data [52]. The observed limitations of live-attenuated vaccines suggest the need to explore alternative vaccination strategies that offer comparable efficacy with reduced adverse effects.
(2) The text construction throughout the results is extremely long. Dividing them into smaller parts could make it easier to read.
Ans. We appreciate the reviewer for the feedback. We have revised the results by breaking them into shorter, more digestible parts to improve clarity and flow.
(3) Epidemiological data needs to be updated. The manuscript cites 41,664 confirmed cases and five deaths, where the authors mention information up to July 2024. Including recent data could show the current epidemiological parameter.
Ans. We thank the reviewer for noting this important point. We have updated the epidemiological data in the manuscript.
Page 1 Line 15-17
Monkeypox virus (MPXV), has caused 148,892 confirmed cases and 341 deaths from 137 countries worldwide, as reported by the World Health Organization (WHO), highlighting the urgent need for effective vaccines to prevent the spread of MPXV.
Page 2 Line 52-56
Since 2022, the global incidence of monkeypox (MPX) disease caused by the virus has dramatically increased worldwide, with over 148,892 laboratory-confirmed infection cases and more than 341 deaths reported by the World Health Organization (WHO) from 137 reporting countries (as updated in WHO website on 31st May 2025.) [6].
Page 21 Line 484-487
Monkeypox is a zoonotic double-stranded DNA virus in the Orthopoxviral genus. Recently, there has been a global outbreak of monkeypox disease, with 148 892 confirmed cases and 341 deaths worldwide according to WHO as updated in WHO website on 31st May 2025 [6].
(4) Although the epitope selection is well explained, it would be useful to include a flowchart summarizing the filtering process (antigenicity, allergenicity, toxicity, etc.). In addition, the rationale for the choice of adjuvants (PADRE, beta-defensin, etc.) could be expanded, highlighting why PADRE was selected as the best candidate.
Ans. We thank the reviewer for this feedback. We have included a flowchart summarizing the filtering process (antigenicity, allergenicity, toxicity, etc.) as per reviewer’s suggestion. We have included the rationale for the choice of adjuvants in this study.
Page 14 Line 439-440
Figure 1. Schematic workflow for designing a broad-spectrum, multi-epitope pan-Mpox vaccine
Page 8 Line 238-250
By incorporating the adjuvants into the vaccine construct, this could improve the immune repsonse. Therefore, in this study we selected five different adjuvants, namely, cholera toxin B subunit (CTB), PADRE, beta-defensin 3, 50S ribosomal protein L7/12, and RS-09, were used in this study. The study showed that CTB can enhance immune responses by binding to GM1 ganglioside receptors on various immune cells [40]. PADRE elicits stronger antibody responses compared to traditional T-cell epitopes [41,42]. Adjuvants like RS09 can improve vaccine efficacy by enhancing, modulating, and prolonging immune responses, while reducing antigen concentration and required immunizations [43]. The 50S Ribosomal L7/L12 can enhance dendritic cell maturation, promote Th1 polarization, and improve T cell-mediated cytotoxicity, leading to better tumor immunotherapy outcomes [44]. Beta-defensins offers several advantages. They enhance both humoral and cell-mediated immune responses, leading to improved vaccine efficacy against various pathogens, including influenza, MERS-CoV, and IBDV [45–47].
(5) Overall, the figures (such as population coverage and molecular docking) are essential, but some captions are too brief. Including more detailed descriptions could help the reader interpret the results without having to search for information in the running text.
Ans. We thank the reviewer for this constructive feedback. In response, we have revised the figure captions-particularly for the population coverage and molecular docking-to include more detailed descriptions. This will help readers interpret the results more easily without needing to refer back to the main text.
(6) Table A1 (in the Appendix) is extensive and difficult to follow. A simplified version or a summary of the most relevant epitopes would be desirable.
Ans. We thank the reviewer for this helpful feedback. In response, we have revised Table A1 to simplify its presentation and improve clarity.
(7) Comparing the results with other recent studies of multi-epitope vaccines for MPXV would enrich the discussion. Please highlight the limitations of the study (such as the reliance on computational predictions and the need for experimental validation) as future research perspectives.
Ans. We thank reviewer for this suggestion, and we have included the limitations of computational analysis and future research perspectives in discussion section.
Page 24 Line 610-619
However, computational analysis is not sufficient to yield accurate results. In vitro characterization and in vivo studies are indispensable for validating the efficacy of broad-spectrum multi-epitope vaccine design.
The critical steps include protein expression in E. coli to confirm its expression as predicted by its physicochemical properties, as well as in vivo studies in model organisms. Furthermore, the potential of this vaccine design can be significantly enhanced by exploring advanced delivery systems such as mRNAs and outer membrane vesicles (OMVs). These additional studies will not only affirm the efficacy and safety of the vaccine but also open new avenues for innovative approaches in vaccine development, offering a robust defense against diverse and evolving viral threats.
(8) Please review the grammar and names throughout the manuscript. Check for typos (e.g., "Vaxijen" instead of "VaxiJen" in some places).
Ans. We appreciate the reviewer raising this point. We have carefully reviewed the manuscript for grammatical errors, typographical issues, and consistency in the use of names throughout. Regarding the specific case of "VaxiJen", we would like to clarify that "VaxiJen" is the correct and official spelling of the tool, as confirmed by its original publications. For reference, the name appears as "VaxiJen" in the following key sources:
- Doytchinova IA, Flower DR. VaxiJen: a server for prediction of protective antigens, tumour antigens and subunit vaccines. BMC Bioinformatics. 2007;8:4.
- Doytchinova IA, Flower DR. Identifying candidate subunit vaccines using an alignment-independent method based on principal amino acid properties. Vaccine. 2007;25:856–866.
- Doytchinova IA, Flower DR. Bioinformatic Approach for Identifying Parasite and Fungal Candidate Subunit Vaccines. Open Vaccines Journal. 2008;1:22–26.
We have ensured that this spelling is used consistently across the manuscript.
(9) Please include a paragraph in the discussion about the general limitations of the methods used.
Ans. Ans. We thank reviewer for this suggestion, and we have included the general limitations of computational analysis.
Page 24 Line 610-612
However, computational analysis is not sufficient to yield accurate results. In vitro characterization and in vivo studies are indispensable for validating the efficacy of broad-spectrum multi-epitope vaccine design.
(10) The authors should review the molecular dynamics simulation part. I consider a 50 ns simulation too short for a conclusion of this size to be published. The authors should extend this simulation to 200 ns and check for stochastic convergence, mainly due to the fact that it has predictions of structures without real experimental data. The authors should include the radius of gyration (Rg) and check that Rg does not change abruptly over time.
Ans. We thank the reviewer for raising this important point. We acknowledge that a longer molecular dynamics simulation, such as 200 ns, would provide deeper insights, particularly in the absence of experimental structural data. However, in this study, we conducted a 50 ns simulation, which we believe still provides meaningful preliminary insights into the stability of the vaccine–TLR complex.
We would also like to clarify that the decision to conduct a 50 ns MD simulation was based on previous studies with similar scope, such as:
- Kaur A, Kumar A, Kumari G, Muduli R, Das M, Kundu R, Mukherjee S, Majumdar T. Rational design and computational evaluation of a multi-epitope vaccine for monkeypox virus: Insights into binding stability and immunological memory. Heliyon. 2024 Aug 30;10(16).
- Tan C, Zhu F, Pan P, Wu A, Li C. Development of multi-epitope vaccines against the monkeypox virus based on envelope proteins using immunoinformatics approaches. Frontiers in Immunology. 2023 Mar 13;14:1112816.
Additionally, we utilized the iMODS server, which provides information on the deformability graph, B-factor graph (RMSD values), eigenvalues, variance plot, covariance map, and elastic network map. Based on the results obtained from the iMODS server, as shown in Figure 9, the TLR-vaccine complex demonstrated low deformability, suggesting structural stability. Collectively, the molecular dynamics simulation results obtained from GROMACS V2024 software and iMODs were consistent and mutually supportive.
Regarding the radius of gyration (Rg), this analysis was performed to assess the structural compactness of the complex. As shown in Figure 10c and described on Page 13, Lines 423–425, the Rg remained substantially stable throughout the simulation period, with no abrupt changes over time.
“Radius of gyration (Rg) was analyzed to determine the structural compactness of the vaccine–TLR complex. As illustrated in Fig. 10c, the overall Rg remained stable throughout the simulation and did not show abrupt fluctuations.”
While we are not in a position to extend the simulation at this stage, we greatly appreciate your valuable suggestion and will strongly consider incorporating longer simulations and convergence checks in our future work.
Round 2
Reviewer 1 Report
Comments and Suggestions for Authors
Thank you for submitting the revised version of your manuscript. I appreciate the efforts you have made to address the prior concerns. Upon review, I find that the revisions have improved the manuscript's clarity and structure.
Reviewer 2 Report
Comments and Suggestions for Authors
Accepted